# A New Approach for Dynamic Stochastic Fractal Search with Fuzzy Logic for Parameter Adaptation

**Marylu L. Lagunes, Oscar Castillo \***, **Fevrier Valdez**, **Jose Soria and Patricia Melin**

Tijuana Institute of Technology, Calzada Tecnologico s/n, Fracc. Tomas Aquino, Tijuana 22379, Mexico; marylu.lara@tectijuana.edu.mx (M.L.L.); fevrier@tectijuana.mx (F.V.); jsoria57@gmail.com (J.S.); pmelin@tectijuana.mx (P.M.)
\* Correspondence: ocastillo@tectijuana.mx

**Abstract:** Stochastic fractal search (SFS) is a novel method inspired by the process of stochastic growth in nature and the use of the fractal mathematical concept. Considering the chaotic stochastic diffusion property, an improved dynamic stochastic fractal search (DSFS) optimization algorithm is presented. The DSFS algorithm was tested with benchmark functions, such as the multimodal, hybrid, and composite functions, to evaluate the performance of the algorithm with dynamic parameter adaptation with type-1 and type-2 fuzzy inference models. The main contribution of the article is the utilization of fuzzy logic in the adaptation of the diffusion parameter in a dynamic fashion. This parameter is in charge of creating new fractal particles, and the diversity and iteration are the input information used in the fuzzy system to control the values of diffusion.

**Keywords:** fractal search; fuzzy logic; parameter adaptation; CEC 2017





## 1. Introduction

Metaheuristic algorithms are applied to optimization problems due to their characteristics that help in searching for the global optimum, while simple heuristics are mostly capable of searching for the local optimum and are not very effective for finding optimal solutions in real problems. The multi-metaheuristic models are those formed by more than two metaheuristics to solve a common optimization problem where each of the metaheuristics is used as an optimization tool. The chosen metaheuristic depends on the problem to be solved; most of the metaheuristic methods have the same inspiration and goal means, and generally, their performance is better than a simple heuristic. All metaheuristic algorithms use certain trade-offs between randomization and local search. These algorithms have two main characteristics, intensification and diversification. Diversification refers to generating solutions to explore the search space on a global scale, and intensification refers to focusing on the search for a local region until the best region is found. The combination of these two features results in global optimization [1]. Stochastic methods have the peculiarity of being characterized by stochastic random variables. In the current literature, several of the most popular stochastic algorithms for optimization can be found, like the genetic algorithm (GA) [2,3], which was inspired by biological evolution, with random genetic combinations and mutations in a chromosome, and is based on the selection, crossover, mutation, and replacement operators. The inspiration for particle swarm optimization (PSO) was natural fish and bird behavior when moving in swarms; each particle moves randomly to find the global optimum, updating its speed and position until finding the best global solution [4]. Tabu search (TS) [5,6] is an iterative method that builds meta-strategies to build a neighborhood and avoids getting trapped in local optima. As mentioned above, these algorithms are the most widely used; for example, in [7], an algorithm for optimization of hybrid particle swarms incorporating chaos is proposed, where the adaptive inertia weight factor (AIWF) is used for enhancing the PSO in balancing, in an efficient way, the diversification and intensification abilities of the algorithm. In this case, the hybridization of PSO with AIWF

and chaos was used to build a chaotic PSO (CPSO) by prudently combining the evolutionary search ability based on the population of the PSO and the behavior of the chaotic search. Furthermore, [8] proposes a new PSO algorithm that relies on chaotic equation maps for parameter adaptation, and this is done through the use of chaotic number generators every time the classical PSO algorithm needs a random number [9]. On the other hand, [10] developed an improved particle swarm optimization (IPSO) algorithm for enhancing the performance of the traditional PSO, which uses a dynamic inertia weight. In [11], the authors present an invasive weed optimization (IWO) metaheuristic, which is a weed colony-based population optimization approach also based on chaos theory. In addition to the improvements to stochastic optimization methods, there are also combinations or hybridizations with fuzzy logic to improve performance or reach a specific solution. For example, in [12], an adaptive fuzzy control approach to control chaotic unified systems was proposed. In addition, in [13–16], a multi-metaheuristic model was developed for the optimal design of fuzzy controllers. An enhancement to the Bat algorithm was carried out in [17] for the dynamic adaptation of parameters. There are different applications using metaheuristics and fuzzy logic for optimization [18,19]. In this article, we focus on testing the efficiency of the dynamic stochastic fractal search (DSFS) method in optimizing unimodal, multimodal, hybrid, and composite functions. First, the stochastic fractal search (SFS) algorithm was considered by analyzing its chaotic stochastic characteristics in the diffusion process, where each of its particles was generated and moved in a random stochastic way. Second, it was detected that the stochastic movement of each particle may not be optimal due to the formation of the fractal not being able to achieve exploration and exploitation of the entire search space. Therefore, diversity was introduced for each iteration to eventually get closer to the global optima by looking for the particles with the best fitness and adapting an inference system for their adjustment. To obtain a comparison of the efficiency of the improved method, 30 functions with different dimensions were evaluated, generating satisfactory results compared to other algorithms. The main contribution of this article is the improvement of the SFS method since the algorithm had a disadvantage in the diffusion parameter because it only uses the Gaussian distribution as its randomness method to compensate for the fact that the particles might not be able to cover all of the search space. Thus, it was decided to add dynamic adjustment to the mentioned parameter to achieve better movement in each of the newly generated fractal particles; this improvement was implemented using type-1 and type-2 fuzzy systems. This was done by making dynamic changes with a controller, which had diversity as Input 1 and iteration as Input 2. Diversification is a charge of spreading the particles throughout the search area for each iteration. As a result, a dynamically adapted method was obtained, which does not stagnate as fast in the optimal local ones, thus reaching the global optimum, and therefore, improving the effectiveness of the dynamic stochastic fractal search (DSFS) method. The motivation for the development of this article was the creation of a stochastic fractal method with dynamic adjustment that does not remain stagnant in the local optimum and achieves reaching the global optimum of the objective function. To comply with this, it was designed with a fuzzy inference system that controls the diffusion of the particles in each iteration through diversity. Applied in this way, the method has less chance of stagnation and premature convergence. The goal was to build an efficient but practical algorithm that works most of the time and is capable of producing good quality solutions for finding the overall optimal solution of real-world problems.

The rest of this article is structured as follows. Section 2 puts forward the stochastic fractal search (SFS) method. Section 3 outlines the proposed dynamic stochastic fractal search (DSFS). In Section 4, a summary of the experimental results is presented. In Section 5, the advantages of the method are highlighted with a discussion of the achieved results. Lastly, in Section 6, conclusions about the modified dynamic stochastic fractal search (DSFS) method are presented.

## 2. Materials and Methods for Stochastic Fractal Search (SFS)

The term "fractal" was used for the first time by Benoit Mandelbrot [20], who described it, in his theory of fractals, as geometric patterns generated in nature. There are some methods to generate fractals, such as systems of iterated functions [21], strange attractors [22], L-systems [23], finite subdivision rules [24], and random fractals [25]. On the part of the generation of random fractals, we can find the Diffusion Limited Aggregation (DLA), which consists of the formation of fractals, starting with an initial particle that is called a "seed" and is usually situated at the origin [26,27]. Then, other particles are randomly generated near the origin, causing diffusion. This diffusion process is carried out with a mathematical algorithm as a random walk, where the diffuser particle adheres to the initial particle. This process is iteratively repeated and stops only when a group of particles is formed. While the group is forming, the probability of a particle getting stuck at the end is incremented with respect to those that reach the interior, forming a cluster with a structure similar to a branch. These branches can shape chaotic stochastic patterns, such as the formation of lightning in nature [28].

In stochastic fractal search (SFS) [29], two important processes occur: the diffusion and the update. In the first one, the particles diffuse near their position to fulfill the intensification property (exploitation); with this, the possibility of finding the global minima is increased, and at the same time, it avoids getting stuck at a local minima. In the second one, a simulation of how one point is updating its position using the positions of other points in the group is made. In this process, the best particle produced from diffusion is the only one that is taken into account, and the remaining particles are eliminated. The equations used in each of the aforementioned processes are explained below. Equation (1) establishes how to generate an initial population for a particular problem based on its constraints:

$$P = LB + \varepsilon * (UP - LB) \tag{1}$$

where the particle population $P$ is randomly produced, considering the problem constraints after setting the lower ($LB$) and the upper ($UB$) limits, and $\varepsilon$ represents a number randomly produced in the range 0–1.

The process of diffusion (exploitation in fractal search) is expressed as follows:

$$GW_1 = \text{Gaussian}\,(\mu_{BP}, \sigma) + (\varepsilon \times BP - \varepsilon' \times P_i) \tag{2}$$

$$GW_2 = \text{Gaussian}\,(\mu_P, \sigma) \tag{3}$$

where $\varepsilon$ and $\varepsilon'$ represent randomly generated numbers in the range of 0–1, $BP$ represents the best position of the point, $i$-th indicates a point $P_i$ and Gaussian represents a normal distribution that randomly generates numbers with a mean $\mu$ and a standard deviation $\sigma$:

$$\sigma = \frac{\log g}{g} \times |P_i - BP| \tag{4}$$

where $\frac{\log g}{g}$ tends to a zero value as $g$ increases.

The update process (representing exploration in a fractal search) is expressed as follows:

$$Pa_i = \frac{rank(P_i)}{N} \tag{5}$$

where $N$ represents the number of particles and $Pa_i$ is the estimated particle probability, whose rank is given by the "rank" function. A classification of the particles is done according to their fitness value. Finally, a probability is assigned to each particle $i$.

$$P_i'(j) = P_x(j) - \varepsilon \times (P_y(j) - P_i(j)) \tag{6}$$

where the augmented component is given by $P_i'(j)$, and $P_x$ and $P_y$ are different points selected from the group in a random fashion. $P_i'$ replaces $P_i$ if it achieves a better fitness value.

$$P_i' = P_i - \varepsilon \times (P_x - BP) \big| \varepsilon' \leq 0.5 \tag{7}$$

$$P_i' = P_i - \varepsilon \times (P_x - P_y) \big| \text{otherwise} \tag{8}$$

Once the first updating stage is finished, the second one is initiated with a ranking of all the points based on Equations (7) and (8). As previously mentioned, if $Pa_i$ is lower than a random $\varepsilon$, the current point, $P_i$, is changed by using the previous equations, in which the $x$ and $y$ indices should be different. Of course, the new $P_i'$ is substituted by $P_i$ if it has a better value.

## 3. Proposed Dynamic Stochastic Fractal Search (DSFS)

As mentioned above, the SFS [30–32] has two important processes: the diffusion method and the updating strategy. In analyzing the diffusion method, we know that a seed particle is initialized, and others are generated and adhere to it through diffusion, which forms a chaotic stochastic fractal branch. By taking this fact into account, the stochastic fractal search (SFS) method has been improved by adding diversity to the diffusion process for the particles in each iteration. In this way, the Gaussian random walk was improved because the particles have more possibilities to exploit the search space and therefore do not stagnate in a suboptimal location. To control the diffusion parameter, a fuzzy inference system was introduced, which dynamically adjusted the diffusion of the particles in a range of 0–1. This fuzzy system for control has two inputs (iteration and diversity) and one output (diffusion), as illustrated in Figures 1 and 2. The implemented fuzzy system has 9 if–then rules, which allow for the expression of the knowledge available about the relationship between the antecedents and consequents. These rules determine the behavior of the fuzzy controller and it is here that the output parameter is emulated. The idea was to have an efficient algorithm that improves the disadvantages of the SFS, which is capable of producing solutions of good quality. In this case, Equations (11) and (12) represent diversity and iteration, respectively.

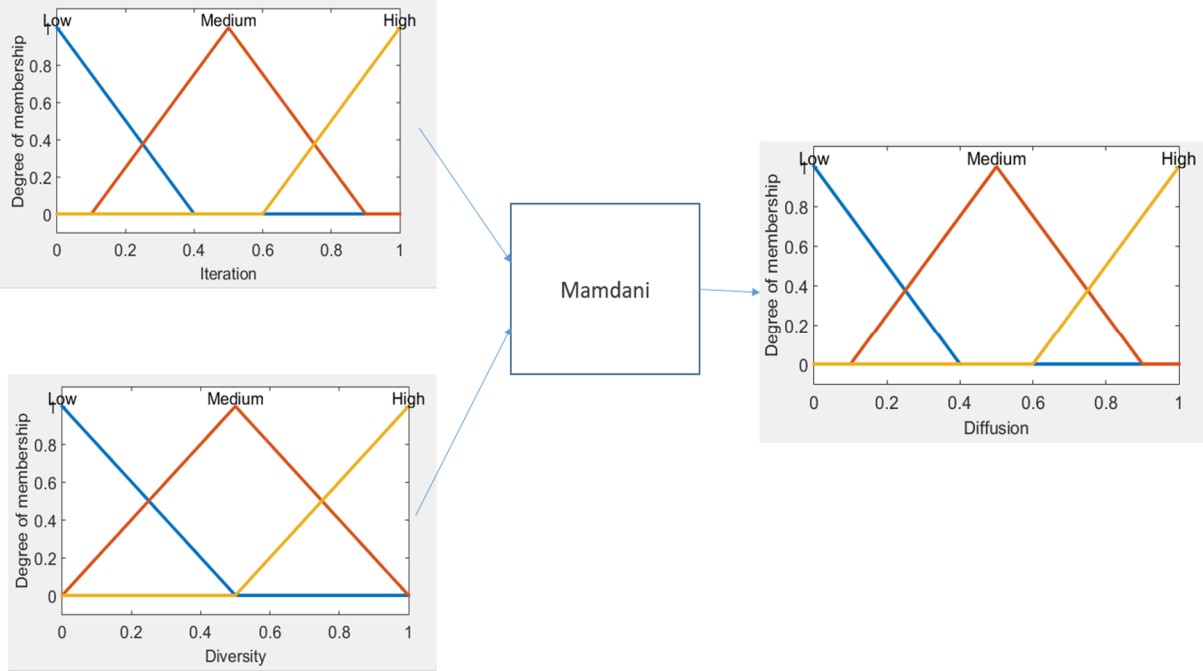

**Figure 1.** Type-1 fuzzy system for controlling diffusion.

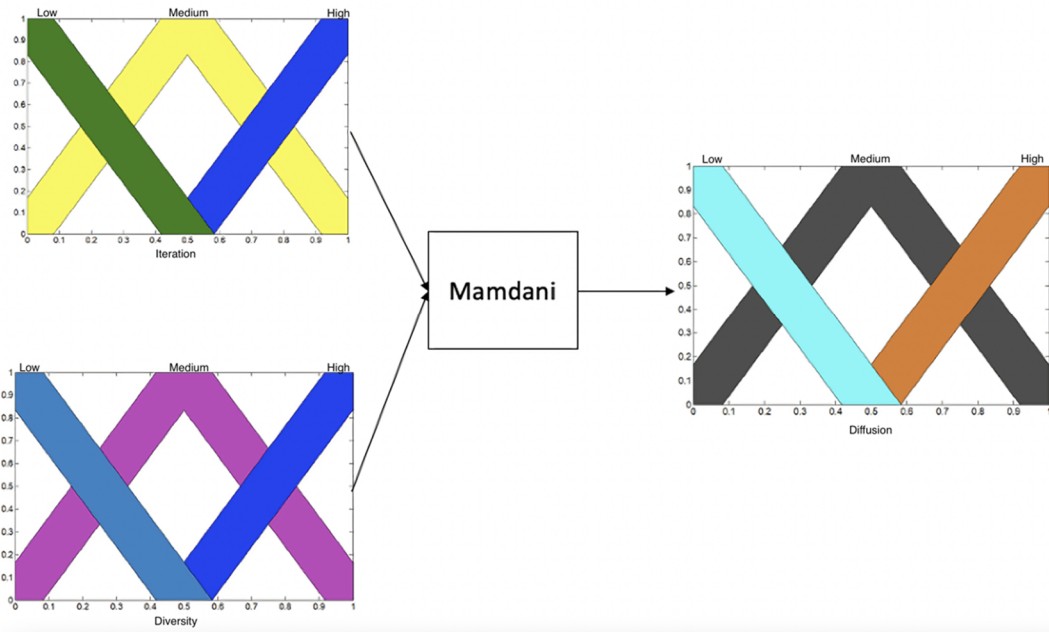

**Figure 2.** Type-2 fuzzy system for controlling the diffusion.

The fuzzy if–then rules are as follows:

If (Iteration is Low) and (Diversity is Low) then (Diffusion is High).
If (Iteration is Low) and (Diversity is Medium) then (Diffusion is Medium).
If (Iteration is Low) and (Diversity is High) then (Diffusion is Medium).
If (Iteration is Medium) and (Diversity is Low) then (Diffusion is Medium).
If (Iteration is Medium) and (Diversity is Medium) then (Diffusion is Medium).
If (Iteration is Medium) and (Diversity is High) then (Diffusion is Medium).
If (Iteration is High) and (Diversity is Low) then (Diffusion is Medium).
If (Iteration is High) and (Diversity is Medium) then (Diffusion is Medium).
If (Iteration is High) and (Diversity is High) then (Diffusion is Low).

Equations (9) and (10) mathematically define the geometrical shape of the triangular functions for type-1 and type-2 fuzzy logic, respectively [33–37].

$$triangular(u; a, b, c) = \begin{cases} 0, & u \leq a \\ \frac{u-a}{b-c}, & a \leq x \leq b \\ \frac{c-x}{c-b}, & b \leq x \leq c \\ 0, & c \leq x \end{cases} \tag{9}$$

$$\widetilde{\mu}(x) = \left[\underline{\mu}(x), \overline{\mu}(x)\right] = \text{itrapatype2}(x, [a_1, b_1, c_1, d_1, a_2, b_2, c_2, d_2, \alpha])$$

where $a_1 < a_2$, $b_1 < b_2$, $c_1 < c_2$, $d_1 < d_2$.

$$\mu_1(x) = \max\left(\min\left(\frac{x-a_1}{b_1-a_1}, 1, \frac{d_1-x}{d_1-c_1}\right), 0\right)$$

$$\mu_2(x) = \max\left(\min\left(\frac{x-a_2}{b_2-a_2}, 1, \frac{d_2-x}{d_2-c_2}\right), 0\right) \tag{10}$$

$$\overline{\mu}(x) = \begin{cases} max(\mu_1(x), \mu_2(x)) & \forall x \notin (b1, c2) \\ 1 & \forall x \in (b1, c2) \end{cases}$$

$$\underline{\mu}(x) = min(\alpha, min(\mu_1(x), \mu_2(x)))$$

$$Iteration = \frac{Current\ Iteration}{Total\ of\ Iterations} \tag{11}$$

$$Diversity\ (S(t)) = \frac{1}{n} \sum_{x=1}^{n} \sqrt{\sum_{y=1}^{D} \left[ P_x(t) - P_y(t) \right]^2} \tag{12}$$

Diversity contributes to the stochastic movement of the particles by having more possibilities of exploiting the entire search space. In this case, in each iteration of the search process, the parameter is dynamically adjusted close to the global optimum, and thus, the diffusion process is improved and we have a more efficient search method.

## 4. Experimental Results

To obtain an understanding of the effectiveness of the proposed dynamic stochastic fractal search (DSFS) method, 30 optimization functions of the Congress on Evolutionary Computing 2017 (CEC 2017) competition [38], summarized in Table 1, were evaluated. In Table 1, we can find several types of functions, such as unimodal, multimodal, hybrid, and composite. To compare the optimization performance of the proposal with respect to other methods, different numbers of dimensions (10, 30, 50, and 100) and different adaptation strategies for the parameters (using type-1 and type-2 fuzzy systems) were used.

**Table 1.** CEC 2017 optimization functions.

| Type of Function | No | Name of Function | fi |
|---|---|---|---|
| Unimodal Functions | 1 | Shifted and Rotated Bent Cigar Function | 100 |
| | 2 | Shifted and Rotated Sum of Different Power Function | 200 |
| | 3 | Shifted and Rotated Zakharov Function | 300 |
| Simple Multimodal Functions | 4 | Shifted and Rotated Rosenbrock's Function | 400 |
| | 5 | Shifted and Rotated Rastrigin's Function | 500 |
| | 6 | Shifted and Rotated Expanded Schaffer's Function | 600 |
| | 7 | Shifted and Rotated Lunacek Bi-Rastrigin Function | 700 |
| | 8 | Shifted and Rotated Non-Continuous Rastrigin's Function | 800 |
| | 9 | Shifted and Rotated Levy Function | 900 |
| | 10 | Shifted and Rotated Schwefel's Function | 1000 |
| Hybrid Functions | 11 | Hybrid Function 1 (N = 3) | 1100 |
| | 12 | Hybrid Function 2 (N = 3) | 1200 |
| | 13 | Hybrid Function 3 (N = 3) | 1300 |
| | 14 | Hybrid Function 4 (N = 4) | 1400 |
| | 15 | Hybrid Function 5 (N = 4) | 1500 |
| | 16 | Hybrid Function 6 (N = 4) | 1600 |
| | 17 | Hybrid Function 6 (N = 5) | 1700 |
| | 18 | Hybrid Function 6 (N = 5) | 1800 |
| | 19 | Hybrid Function 6 (N = 5) | 1900 |
| | 20 | Hybrid Function 6 (N = 6) | 2000 |
| | 21 | Composition Function 1 (N = 3) | 2100 |
| Composition Functions | 22 | Composition Function 2 (N = 3) | 2200 |
| | 23 | Composition Function 3 (N = 4) | 2300 |
| | 24 | Composition Function 4 (N = 4) | 2400 |
| | 25 | Composition Function 5 (N = 5) | 2500 |
| | 26 | Composition Function 6 (N = 3) | 2600 |
| | 27 | Composition Function 7 (N = 6) | 2700 |
| | 28 | Composition Function 8 (N = 3) | 2800 |
| | 29 | Composition Function 9 (N = 3) | 2900 |
| | 30 | Composition Function 10 (N = 3) | 3000 |

To evaluate the proposed method, the CEC 2017 benchmark mathematical functions that have been widely used in the literature [39] were considered in the tests. Table 2 shows, at the top, the number of dimensions used for each function. Column 1 shows the function that is given as f1, f2, f3, f4, . . . , f30, Column 2, represented as "fi" shows the global optimum (minimum) of each function. The next two columns of Table 2 contain the obtained results by the dynamic stochastic fractal search (DSFS) algorithm using type-1 fuzzy logic. The observable results are the average and standard deviation of each of the functions. Finally, Columns 5 and 6 show the mean and standard deviations, respectively, of the functions when using type-2 fuzzy logic for the adaptation of the parameter values in DSFS.

**Table 2.** Dynamic Stochastic Fractal Search (DSFS) results with 10 dimensions.

| Dynamic Stochastic Fractal Search (DSFS) with 10 Dimensions | | | | | |
|---|---|---|---|---|---|
| | | Type-1 Fuzzy Logic | | Type-2 Fuzzy Logic | |
| Function | fi | Main | Std | Main | Std |
| f1 | 100 | $1.01 \times 10^2$ | $4.87 \times 10^{-1}$ | $1.01 \times 10^2$ | $7.85 \times 10^{-1}$ |
| f2 | 200 | $2.00 \times 10^2$ | 0.00 | $2.00 \times 10^2$ | 0.00 |
| f3 | 300 | $3.00 \times 10^2$ | $3.73 \times 10^{-6}$ | $3.00 \times 10^2$ | $5.04 \times 10^{-6}$ |
| f4 | 400 | $4.01 \times 10^2$ | $7.81 \times 10^{-1}$ | $4.00 \times 10^2$ | $6.19 \times 10^{-1}$ |
| f5 | 500 | $5.06 \times 10^2$ | 2.04 | $5.07 \times 10^2$ | 2.64 |
| f6 | 600 | $6.00 \times 10^2$ | $7.21 \times 10^{-8}$ | $6.00 \times 10^2$ | $6.24 \times 10^{-8}$ |
| f7 | 700 | $7.19 \times 10^2$ | 3.45 | $7.20 \times 10^2$ | 3.32 |
| f8 | 800 | $8.07 \times 10^2$ | 3.04 | $8.07 \times 10^2$ | 2.44 |
| f9 | 900 | $9.00 \times 10^2$ | 0.00 | $9.00 \times 10^2$ | 0.00 |
| f10 | 1000 | $1.40 \times 10^3$ | $1.70 \times 10^2$ | $1.34 \times 10^3$ | $1.31 \times 10^2$ |
| f11 | 1100 | $1.10 \times 10^3$ | $9.31 \times 10^{-1}$ | $1.10 \times 10^3$ | 1.00 |
| f12 | 1200 | $1.50 \times 10^3$ | $1.04 \times 10^2$ | $1.52 \times 10^3$ | $1.02 \times 10^2$ |
| f13 | 1300 | $1.31 \times 10^3$ | 4.03 | $1.31 \times 10^3$ | 4.14 |
| f14 | 1400 | $1.40 \times 10^3$ | 1.69 | $1.40 \times 10^3$ | 2.01 |
| f15 | 1500 | $1.50 \times 10^3$ | $7.24 \times 10^{-1}$ | $1.50 \times 10^3$ | $8.88 \times 10^{-1}$ |
| f16 | 1600 | $1.60 \times 10^3$ | $2.79 \times 10^{-1}$ | $1.60 \times 10^3$ | $3.64 \times 10^{-1}$ |
| f17 | 1700 | $1.70 \times 10^3$ | 2.68 | $1.71 \times 10^3$ | 4.41 |
| f18 | 1800 | $1.81 \times 10^3$ | 2.56 | $1.81 \times 10^3$ | 2.57 |
| f19 | 1900 | $1.90 \times 10^3$ | $4.15 \times 10^{-1}$ | $1.90 \times 10^3$ | $4.45 \times 10^{-1}$ |
| f20 | 2000 | $2.00 \times 10^3$ | $1.39 \times 10^{-1}$ | $2.00 \times 10^3$ | $4.73 \times 10^{-3}$ |
| f21 | 2100 | $2.23 \times 10^3$ | $5.08 \times 10$ | $2.24 \times 10^3$ | $5.40 \times 10$ |
| f22 | 2200 | $2.29 \times 10^3$ | $2.17 \times 10$ | $2.28 \times 10^3$ | $3.93 \times 10$ |
| f23 | 2300 | $2.61 \times 10^3$ | 2.80 | $2.60 \times 10^3$ | $5.61 \times 10$ |
| f24 | 2400 | $2.61 \times 10^3$ | $1.22 \times 10^2$ | $2.63 \times 10^3$ | $1.14 \times 10^2$ |
| f25 | 2500 | $2.90 \times 10^3$ | $1.08 \times 10$ | $2.90 \times 10^3$ | $1.15 \times 10$ |
| f26 | 2600 | $2.90 \times 10^3$ | $2.15 \times 10^{-10}$ | $2.90 \times 10^3$ | $3.02 \times 10^{-10}$ |
| f27 | 2700 | $3.09 \times 10^3$ | 1.97 | $3.09 \times 10^3$ | 2.14 |
| f28 | 2800 | $3.09 \times 10^3$ | $4.20 \times 10$ | $3.11 \times 10^3$ | $5.69 \times 10$ |
| f29 | 2900 | $3.16 \times 10^3$ | $1.01 \times 10$ | $3.16 \times 10^3$ | 9.65 |
| f30 | 3000 | $3.56 \times 10^3$ | $2.30 \times 10^2$ | $3.55 \times 10^3$ | $1.19 \times 10^2$ |

As can be seen, the results using type-1 and type-2 fuzzy systems did not have a significant visible difference using 10 dimensions. Even so, the results obtained were good on average because most reached the optimal global of the functions. In Table 3, Row 3, the first column shows the number of the function that is being evaluated. In Column 2, the optimum of each function is observed. In Column 3, the average results of each of the functions are given, and Column 4 shows the standard deviation; these two were

obtained with type-1 fuzzy logic. Finally, the last columns illustrate the results in average and standard deviation, respectively, with type-2 fuzzy logic. It can be appreciated that the difference between these results was minimal, even though these results were obtained with experimentation using 30 dimensions. With this number of dimensions, the algorithm has a wider search space. In addition, the functions that were optimized are complex. Despite this situation, the method showed that it was approaching the optimum of each function, and the values obtained with both types of fuzzy logic were also very close.

**Table 3.** Dynamic Stochastic Fractal Search (DSFS) results with 30 dimensions.

| Dynamic Stochastic Fractal Search (DFSF) with 30 Dimensions | | | | | |
|---|---|---|---|---|---|
| | | Type-1 Fuzzy Logic | | Type-2 Fuzzy Logic | |
| Function | fi | Main | Std | Main | Std |
| f1 | 100 | $3.49 \times 10^3$ | $2.76 \times 10^3$ | $3.24 \times 10^3$ | $2.74 \times 10^3$ |
| f2 | 200 | $3.06 \times 10^{16}$ | $7.11 \times 10^{16}$ | $7.59 \times 10^{17}$ | $3.98 \times 10^{18}$ |
| f3 | 300 | $8.40 \times 10^3$ | $3.95 \times 10^3$ | $8.79 \times 10^3$ | $4.60 \times 10^3$ |
| f4 | 400 | $4.87 \times 10^2$ | $3.56 \times 10$ | $4.80 \times 10^2$ | $3.39 \times 10$ |
| f5 | 500 | $6.11 \times 10^2$ | $2.27 \times 10$ | $6.00 \times 10^2$ | $2.02 \times 10$ |
| f6 | 600 | $6.00 \times 10^2$ | $1.36 \times 10^{-2}$ | $6.00 \times 10^2$ | $1.13 \times 10^{-2}$ |
| f7 | 700 | $8.53 \times 10^2$ | $1.68 \times 10$ | $8.60 \times 10^2$ | $1.70 \times 10$ |
| f8 | 800 | $9.05 \times 10^2$ | $2.12 \times 10$ | $9.06 \times 10^2$ | $2.27 \times 10$ |
| f9 | 900 | $9.01 \times 10^2$ | $6.89 \times 10^{-1}$ | $9.04 \times 10^2$ | $1.44 \times 10$ |
| f10 | 1000 | $6.21 \times 10^3$ | $6.89 \times 10^{-1}$ | $6.05 \times 10^3$ | $5.03 \times 10^2$ |
| f11 | 1100 | $1.19 \times 10^3$ | $2.31 \times 10$ | $1.19 \times 10^3$ | $2.90 \times 10$ |
| f12 | 1200 | $1.56 \times 10^5$ | $1.03 \times 10^5$ | $1.81 \times 10^5$ | $1.48 \times 10^5$ |
| f13 | 1300 | $3.56 \times 10^3$ | $9.09 \times 10^2$ | $4.01 \times 10^3$ | $1.00 \times 10^3$ |
| f14 | 1400 | $1.50 \times 10^3$ | $1.03 \times 10$ | $1.50 \times 10^3$ | $1.06 \times 10$ |
| f15 | 1500 | $1.70 \times 10^3$ | $4.43 \times 10$ | $1.70 \times 10^3$ | $3.95 \times 10$ |
| f16 | 1600 | $2.45 \times 10^3$ | $2.66 \times 10^2$ | $2.47 \times 10^3$ | $2.14 \times 10^2$ |
| f17 | 1700 | $1.85 \times 10^3$ | $6.65 \times 10$ | $1.86 \times 10^3$ | $8.06 \times 10$ |
| f18 | 1800 | $2.87 \times 10^3$ | $6.86 \times 10^2$ | $2.69 \times 10^3$ | $3.69 \times 10^2$ |
| f19 | 1900 | $1.99 \times 10^3$ | $1.88 \times 10$ | $1.99 \times 10^3$ | $1.87 \times 10$ |
| f20 | 2000 | $2.39 \times 10^3$ | $2.58 \times 10$ | $2.22 \times 10^3$ | $1.12 \times 10^2$ |
| f21 | 2100 | $2.40 \times 10^3$ | $2.28 \times 10$ | $2.39 \times 10^3$ | $2.51 \times 10$ |
| f22 | 2200 | $2.30 \times 10^3$ | $1.50 \times 10^{-2}$ | $2.30 \times 10^3$ | $7.44 \times 10^{-3}$ |
| f23 | 2300 | $2.74 \times 10^3$ | $2.86 \times 10$ | $2.73 \times 10^3$ | $2.26 \times 10$ |
| f24 | 2400 | $2.91 \times 10^3$ | $3.37 \times 10$ | $2.90 \times 10^3$ | $3.04 \times 10$ |
| f25 | 2500 | $2.89 \times 10^3$ | $1.73$ | $2.89 \times 10^3$ | $9.76 \times 10^{-1}$ |
| f26 | 2600 | $4.28 \times 10^3$ | $5.50 \times 10^2$ | $4.22 \times 10^3$ | $6.59 \times 10^2$ |
| f27 | 2700 | $3.22 \times 10^3$ | $8.08$ | $3.22 \times 10^3$ | $7.37$ |
| f28 | 2800 | $3.21 \times 10^3$ | $1.25 \times 10$ | $3.21 \times 10^3$ | $1.27 \times 10$ |
| f29 | 2900 | $3.63 \times 10^3$ | $1.06 \times 10^2$ | $3.59 \times 10^3$ | $1.13 \times 10^2$ |
| f30 | 3000 | $1.32 \times 10^4$ | $3.29 \times 10^3$ | $1.51 \times 10^4$ | $5.55 \times 10^3$ |

Multimodal functions are more difficult to optimize than unimodal ones due to the complexity that they represent because the algorithms must escape or avoid local optima and arrive at the global optimal solution. In this study, not only unimodal and multimodal functions were optimized but also hybrid and complex functions were optimized using different values of dimensions, as can be seen in Tables 4 and 5, with 50 and 100 dimensions, respectively. In addition, the values obtained with the variants using type-1 and type-2 fuzzy systems for the adaptation of parameters show that the method had some degree of difficulty in reaching the global optimum. Even so, they provided good approximation values, showing that the improved method is efficient in optimization tasks. The explanation of the rows and columns of Tables 4 and 5 is analogous to the previously mentioned description for Table 3.

**Table 4.** Dynamic Stochastic Fractal Search (DSFS) results with 50 dimensions.

| Dynamic Stochastic Fractal Search (DSFS) with 50 Dimensions | | | | | |
|---|---|---|---|---|---|
| | | Type-1 Fuzzy Logic | | Type-2 Fuzzy Logic | |
| Function | fi | Main | Std | Main | Std |
| f1 | 100 | $9.00 \times 10^4$ | $4.30 \times 10^4$ | $8.42 \times 10^4$ | $6.69 \times 10^4$ |
| f2 | 200 | $1.20 \times 10^{40}$ | $4.53 \times 10^{40}$ | $5.08 \times 10^{39}$ | $1.52 \times 10^{40}$ |
| f3 | 300 | $6.01 \times 10^4$ | $9.72 \times 10^3$ | $6.28 \times 10^4$ | $1.39 \times 10^4$ |
| f4 | 400 | $5.72 \times 10^2$ | $4.05 \times 10$ | $5.69 \times 10^2$ | $4.26 \times 10$ |
| f5 | 500 | $7.68 \times 10^2$ | $4.35 \times 10$ | $7.69 \times 10^2$ | $4.59 \times 10$ |
| f6 | 600 | $6.01 \times 10^2$ | $1.35 \times 10^{-1}$ | $6.01 \times 10^2$ | $1.55 \times 10^{-1}$ |
| f7 | 700 | $1.05 \times 10^3$ | $3.18 \times 10$ | $1.06 \times 10^3$ | $2.61 \times 10$ |
| f8 | 800 | $1.06 \times 10^3$ | $3.69 \times 10$ | $1.06 \times 10^3$ | $4.42 \times 10$ |
| f9 | 900 | $1.13 \times 10^3$ | $1.49 \times 10^2$ | $1.14 \times 10^3$ | $1.22 \times 10^2$ |
| f10 | 1000 | $1.13 \times 10^4$ | $4.84 \times 10^2$ | $1.12 \times 10^4$ | $6.53 \times 10^2$ |
| f11 | 1100 | $1.36 \times 10^3$ | $3.00 \times 10$ | $1.36 \times 10^3$ | $4.30 \times 10$ |
| f12 | 1200 | $3.54 \times 10^6$ | $1.42 \times 10^6$ | $3.47 \times 10^6$ | $1.96 \times 10^6$ |
| f13 | 1300 | $1.48 \times 10^4$ | $1.17 \times 10^4$ | $1.67 \times 10^4$ | $1.36 \times 10^4$ |
| f14 | 1400 | $1.50 \times 10^3$ | $1.03 \times 10$ | $1.80 \times 10^3$ | $8.36 \times 10$ |
| f15 | 1500 | $4.14 \times 10^3$ | $1.54 \times 10^3$ | $3.75 \times 10^3$ | $8.43 \times 10^2$ |
| f16 | 1600 | $3.52 \times 10^3$ | $3.87 \times 10^2$ | $3.58 \times 10^3$ | $4.80 \times 10^2$ |
| f17 | 1700 | $3.05 \times 10^3$ | $2.37 \times 10^2$ | $3.04 \times 10^3$ | $2.56 \times 10^2$ |
| f18 | 1800 | $5.42 \times 10^4$ | $3.34 \times 10^4$ | $5.39 \times 10^4$ | $3.20 \times 10^4$ |
| f19 | 1900 | $7.10 \times 10^3$ | $4.05 \times 10^3$ | $7.73 \times 10^3$ | $5.45 \times 10^3$ |
| f20 | 2000 | $3.10 \times 10^3$ | $2.16 \times 10^2$ | $3.14 \times 10^3$ | $2.36 \times 10^2$ |
| f21 | 2100 | $2.55 \times 10^3$ | $4.55 \times 10$ | $2.55 \times 10^3$ | $4.99 \times 10$ |
| f22 | 2200 | $1.10 \times 10^4$ | $4.12 \times 10^3$ | $1.23 \times 10^4$ | $3.40 \times 10^3$ |
| f23 | 2300 | $3.00 \times 10^3$ | $4.43 \times 10$ | $2.98 \times 10^3$ | $4.91 \times 10$ |
| f24 | 2400 | $3.14 \times 10^3$ | $5.98 \times 10$ | $3.14 \times 10^3$ | $5.85 \times 10$ |
| f25 | 2500 | $3.07 \times 10^3$ | $2.52 \times 10$ | $3.08 \times 10^3$ | $2.22 \times 10$ |
| f26 | 2600 | $6.07 \times 10^3$ | $5.92 \times 10^2$ | $6.16 \times 10^3$ | $4.77 \times 10^2$ |
| f27 | 2700 | $3.41 \times 10^3$ | $4.40 \times 10$ | $3.40 \times 10^3$ | $4.09 \times 10$ |
| f28 | 2800 | $3.36 \times 10^3$ | $3.36 \times 10$ | $3.35 \times 10^3$ | $3.44 \times 10$ |
| f29 | 2900 | $4.15 \times 10^3$ | $2.71 \times 10^2$ | $4.17 \times 10^3$ | $2.55 \times 10^2$ |
| f30 | 3000 | $3.25 \times 10^6$ | $6.84 \times 10^5$ | $3.24 \times 10^6$ | $8.20 \times 10^5$ |

**Table 5.** Dynamic Stochastic Fractal Search (DSFS) results with 100 dimensions.

| Dynamic Stochastic Fractal Search DSFS with 100 Dimensions | | | | | |
|---|---|---|---|---|---|
| | | Type-1 Fuzzy Logic | | Type-2 Fuzzy Logic | |
| Function | fi | Main | Std | Main | Std |
| f1 | 100 | $1.15 \times 10^8$ | $4.36 \times 10^7$ | $1.08 \times 10^8$ | $3.31 \times 10^7$ |
| f2 | 200 | $1.34 \times 10^{108}$ | $8.85 \times 10^{108}$ | $1.14 \times 10^{111}$ | $5.81 \times 10^{111}$ |
| f3 | 300 | $2.53 \times 10^5$ | $2.78 \times 10^4$ | $2.56 \times 10^5$ | $2.78 \times 10^4$ |
| f4 | 400 | $9.23 \times 10^2$ | $4.74 \times 10$ | $9.37 \times 10^2$ | $6.10 \times 10$ |
| f5 | 500 | $1.29 \times 10^3$ | $9.15 \times 10$ | $1.29 \times 10^3$ | $7.11 \times 10$ |
| f6 | 600 | $6.07 \times 10^2$ | $1.33$ | $6.07 \times 10^2$ | $1.02$ |
| f7 | 700 | $1.72 \times 10^3$ | $5.61 \times 10$ | $1.73 \times 10^3$ | $5.30 \times 10$ |
| f8 | 800 | $1.58 \times 10^3$ | $7.70 \times 10$ | $1.59 \times 10^3$ | $8.69 \times 10$ |
| f9 | 900 | $1.19 \times 10^4$ | $3.07 \times 10^3$ | $1.28 \times 10^4$ | $4.21 \times 10^3$ |
| f10 | 1000 | $2.71 \times 10^4$ | $1.04 \times 10^3$ | $2.72 \times 10^4$ | $9.91 \times 10^2$ |
| f11 | 1100 | $1.62 \times 10^4$ | $3.86 \times 10^3$ | $1.63 \times 10^4$ | $4.35 \times 10^3$ |
| f12 | 1200 | $6.66 \times 10^7$ | $2.09 \times 10^7$ | $7.05 \times 10^7$ | $1.74 \times 10^7$ |
| f13 | 1300 | $5.66 \times 10^3$ | $1.80 \times 10^3$ | $6.16 \times 10^3$ | $2.89 \times 10^3$ |
| f14 | 1400 | $3.36 \times 10^5$ | $2.50 \times 10^5$ | $2.36 \times 10^5$ | $1.44 \times 10^5$ |
| f15 | 1500 | $4.46 \times 10^3$ | $4.66 \times 10^3$ | $4.39 \times 10^3$ | $3.13 \times 10^3$ |

**Table 5.** *Cont.*

| | Dynamic Stochastic Fractal Search DSFS with 100 Dimensions | | | |
|---|---|---|---|---|
| f16 | 1600 | $7.97 \times 10^3$ | $8.58 \times 10^2$ | $7.70 \times 10^3$ | $7.93 \times 10^2$ |
| f17 | 1700 | $6.04 \times 10^3$ | $3.39 \times 10^2$ | $5.90 \times 10^3$ | $5.76 \times 10^2$ |
| f18 | 1800 | $5.97 \times 10^5$ | $3.86 \times 10^5$ | $6.45 \times 10^5$ | $3.99 \times 10^5$ |
| f19 | 1900 | $3.60 \times 10^3$ | $1.65 \times 10^3$ | $3.49 \times 10^3$ | $1.56 \times 10^3$ |
| f20 | 2000 | $2.18 \times 10^3$ | $8.28 \times 10$ | $6.27 \times 10^3$ | $4.30 \times 10^2$ |
| f21 | 2100 | $3.11 \times 10^3$ | $6.75 \times 10$ | $3.11 \times 10^3$ | $6.06 \times 10$ |
| f22 | 2200 | $2.96 \times 10^4$ | $8.67 \times 10^2$ | $2.97 \times 10^4$ | $8.23 \times 10^2$ |
| f23 | 2300 | $3.59 \times 10^3$ | $5.69 \times 10$ | $3.57 \times 10^3$ | $7.87 \times 10$ |
| f24 | 2400 | $4.09 \times 10^3$ | $9.76 \times 10$ | $4.08 \times 10^3$ | $1.19 \times 10^2$ |
| f25 | 2500 | $3.63 \times 10^3$ | $6.17 \times 10$ | $3.63 \times 10^3$ | $5.46 \times 10$ |
| f26 | 2600 | $1.41 \times 10^4$ | $1.04 \times 10^3$ | $1.42 \times 10^4$ | $8.28 \times 10^2$ |
| f27 | 2700 | $3.72 \times 10^3$ | $7.06 \times 10$ | $3.73 \times 10^3$ | $5.89 \times 10$ |
| f28 | 2800 | $3.96 \times 10^3$ | $1.11 \times 10^2$ | $3.96 \times 10^3$ | $1.35 \times 10^2$ |
| f29 | 2900 | $7.96 \times 10^3$ | $5.34 \times 10^2$ | $7.96 \times 10^3$ | $5.11 \times 10^2$ |
| f30 | 3000 | $3.04 \times 10^5$ | $1.27 \times 10^5$ | $3.18 \times 10^5$ | $1.68 \times 10^5$ |

## 5. Discussion of Results

In the literature, we can find the hybrid firefly and particle swarm optimization algorithm for solving expensive computational problems [40], which was also used for the optimization of the CEC 2017 functions, and the performance of which was compared to another four optimization methods. The comparison was done in [40] with the combination of the hybrid firefly algorithm and particles swarm optimization (HFPSO), as well as with the Firefly Algorithm (FA), particle swarm optimization (PSO), Hybrid PSO and Firefly Algorithm (HPSOFF), and Hybrid Firefly and PSO (FFPSO). For this reason, we decided to consider it as a reference for comparison to test the efficacy of the dynamic stochastic fractal search (DFSF) method. The experimentation was carried out with the following specifications: 20 independent runs, and for each case, the maximum number of evaluations of 500 was used for 10 dimensions (10D), and 1500 for 30 dimensions (30D).

The combination of the Firefly Algorithm and Particle Swarm Optimization (HFPSO) was the one that generated the best results in the comparison that was made in the reference article [40], with respect to the other four optimization algorithms. For this reason, in Tables 6 and 7, only the results of the HFPSO are compared against the proposed DSFS with both types of fuzzy logic. The experimentation of the CEC 2017 functions with 10 and 30 dimensions, respectively, show us that the dynamic stochastic fractal search method obtained, on average, better results in finding the global solutions of the functions. As previously mentioned, the results of the dynamic stochastic fractal search with type-1 and type-2 fuzzy systems were very close. Therefore, in comparison with HFPSO, better overall results were also obtained for each function. Figures 3 and 4 graphically illustrate a comparison of the proposed DSFS against the HFPSO for 10 and 30 dimensions, respectively.

**Table 6.** HFPSO vs. DSFS results with 10 dimensions.

| | | HFPSO [40] | | DSFS Type-1 Fuzzy Logic | | DSFS Type-2 Fuzzy Logic | |
|---|---|---|---|---|---|---|---|
| Function | fi | Mean | Std | Mean | Std | Mean | Std |
| f1 | 100 | $9.81 \times 10^8$ | $1.01 \times 10^2$ | $1.01 \times 10^2$ | $4.87 \times 10^{-1}$ | $1.01 \times 10^2$ | $7.85 \times 10^{-1}$ |
| f2 | 200 | $4.91 \times 10^8$ | $2.00 \times 10^2$ | $2.00 \times 10^2$ | 0.00 | $2.00 \times 10^2$ | 0.00 |
| f3 | 300 | $5.96 \times 10^3$ | $3.00 \times 10^2$ | $3.00 \times 10^2$ | $3.73 \times 10^{-6}$ | $3.00 \times 10^2$ | $5.04 \times 10^{-6}$ |
| f4 | 400 | $4.55 \times 10$ | $4.01 \times 10^2$ | $4.01 \times 10^2$ | $7.81 \times 10^{-1}$ | $4.00 \times 10^2$ | $6.19 \times 10^{-1}$ |
| f5 | 500 | $1.84 \times 10$ | $5.06 \times 10^2$ | $5.06 \times 10^2$ | 2.04 | $5.07 \times 10^2$ | 2.64 |
| f6 | 600 | $1.35 \times 10$ | $6.00 \times 10^2$ | $6.00 \times 10^2$ | $7.21 \times 10^{-8}$ | $6.00 \times 10^2$ | $6.24 \times 10^{-8}$ |
| f7 | 700 | $1.73 \times 10$ | $7.19 \times 10^2$ | $7.19 \times 10^2$ | 3.45 | $7.20 \times 10^2$ | 3.32 |

**Table 6.** *Cont.*

| Function | fi | HFPSO [40] | | DSFS Type-1 Fuzzy Logic | | DSFS Type-2 Fuzzy Logic | |
| | | Mean | Std | Mean | Std | Mean | Std |
|---|---|---|---|---|---|---|---|
| f8 | 800 | $1.44 \times 10$ | $8.07 \times 10^2$ | $8.07 \times 10^2$ | 3.04 | $8.07 \times 10^2$ | 2.44 |
| f9 | 900 | $3.07 \times 10^2$ | $9.00 \times 10^2$ | $9.00 \times 10^2$ | 0.00 | $9.00 \times 10^2$ | 0.00 |
| f10 | 1000 | $3.79 \times 10^2$ | $1.40 \times 10^3$ | $1.40 \times 10^3$ | $1.70 \times 10^2$ | $1.34 \times 10^3$ | $1.31 \times 10^2$ |
| f11 | 1100 | $5.24 \times 10$ | $1.10 \times 10^3$ | $1.10 \times 10^3$ | $9.31 \times 10^{-1}$ | $1.10 \times 10^3$ | 1.00 |
| f12 | 1200 | $4.13 \times 10^6$ | $1.50 \times 10^3$ | $1.50 \times 10^3$ | $1.04 \times 10^2$ | $1.52 \times 10^3$ | $1.02 \times 10^2$ |
| f13 | 1300 | $7.68 \times 10^3$ | $1.31 \times 10^3$ | $1.31 \times 10^3$ | 4.03 | $1.31 \times 10^3$ | 4.14 |
| f14 | 1400 | $4.18 \times 10^3$ | $1.40 \times 10^3$ | $1.40 \times 10^3$ | 1.69 | $1.40 \times 10^3$ | 2.01 |
| f15 | 1500 | $2.37 \times 10^4$ | $1.50 \times 10^3$ | $1.50 \times 10^3$ | $7.24 \times 10^{-1}$ | $1.50 \times 10^3$ | $8.88 \times 10^{-1}$ |
| f16 | 1600 | $1.59 \times 10^2$ | $1.60 \times 10^3$ | $1.60 \times 10^3$ | $2.79 \times 10^{-1}$ | $1.60 \times 10^3$ | $3.64 \times 10^{-1}$ |
| f17 | 1700 | $8.40 \times 10$ | $1.70 \times 10^3$ | $1.70 \times 10^3$ | 2.68 | $1.71 \times 10^3$ | 4.41 |
| f18 | 1800 | $1.79 \times 10^4$ | $1.81 \times 10^3$ | $1.81 \times 10^3$ | 2.56 | $1.81 \times 10^3$ | 2.57 |
| f19 | 1900 | $3.83 \times 10^4$ | $1.90 \times 10^3$ | $1.90 \times 10^3$ | $4.15 \times 10^{-1}$ | $1.90 \times 10^3$ | $4.45 \times 10^{-1}$ |
| f20 | 2000 | $1.08 \times 10^2$ | $2.00 \times 10^3$ | $2.00 \times 10^3$ | $1.39 \times 10^{-1}$ | $2.00 \times 10^3$ | $4.73 \times 10^{-3}$ |
| f21 | 2100 | $4.78 \times 10$ | $2.23 \times 10^3$ | $2.23 \times 10^3$ | $5.08 \times 10$ | $2.24 \times 10^3$ | $5.40 \times 10$ |
| f22 | 2200 | $5.88 \times 10^2$ | $2.29 \times 10^3$ | $2.29 \times 10^3$ | $2.17 \times 10$ | $2.28 \times 10^3$ | $3.93 \times 10$ |
| f23 | 2300 | $2.87 \times 10$ | $2.61 \times 10^3$ | $2.61 \times 10^3$ | 2.80 | $2.60 \times 10^3$ | $5.61 \times 10$ |
| f24 | 2400 | $1.47 \times 10^2$ | $2.61 \times 10^3$ | $2.61 \times 10^3$ | $1.22 \times 10^2$ | $2.63 \times 10^3$ | $1.14 \times 10^2$ |
| f25 | 2500 | $5.02 \times 10$ | $2.90 \times 10^3$ | $2.90 \times 10^3$ | $1.08 \times 10$ | $2.90 \times 10^3$ | $1.15 \times 10$ |
| f26 | 2600 | $3.42 \times 10^2$ | $2.90 \times 10^3$ | $2.90 \times 10^3$ | $2.15 \times 10^{-10}$ | $2.90 \times 10^3$ | $3.02 \times 10^{-10}$ |
| f27 | 2700 | $3.94 \times 10$ | $3.09 \times 10^3$ | $3.09 \times 10^3$ | 1.97 | $3.09 \times 10^3$ | 2.14 |
| f28 | 2800 | $1.08 \times 10^2$ | $3.09 \times 10^3$ | $3.09 \times 10^3$ | $4.20 \times 10$ | $3.11 \times 10^3$ | $5.69 \times 10$ |
| f29 | 2900 | $9.40 \times 10$ | $3.16 \times 10^3$ | $3.16 \times 10^3$ | $1.01 \times 10$ | $3.16 \times 10^3$ | 9.65 |
| f30 | 3000 | $3.75 \times 10^6$ | $3.56 \times 10^3$ | $3.56 \times 10^3$ | $2.30 \times 10^2$ | $3.55 \times 10^3$ | $1.19 \times 10^2$ |

**Table 7.** HFPSO vs. DSFS results with 30 dimensions.

| Function | fi | HFPSO [40] | | DSFS Type-1 Fuzzy Logic | | DSFS Type-2 Fuzzy Logic | |
| | | Mean | Std | Mean | Std | Mean | Std |
|---|---|---|---|---|---|---|---|
| f1 | 100 | $9.81 \times 10^8$ | $1.01 \times 10^2$ | $3.49 \times 10^3$ | $2.76 \times 10^3$ | $3.24 \times 10^3$ | $2.74 \times 10^3$ |
| f2 | 200 | $4.91 \times 10^8$ | $2.00 \times 10^2$ | $3.06 \times 10^{16}$ | $7.11 \times 10^{16}$ | $7.59 \times 10^{17}$ | $3.98 \times 10^{18}$ |
| f3 | 300 | $5.96 \times 10^3$ | $3.00 \times 10^2$ | $8.40 \times 10^3$ | $3.95 \times 10^3$ | $8.79 \times 10^3$ | $4.60 \times 10^3$ |
| f4 | 400 | $4.55 \times 10$ | $4.01 \times 10^2$ | $4.87 \times 10^2$ | $3.56 \times 10$ | $4.80 \times 10^2$ | $3.39 \times 10$ |
| f5 | 500 | $1.84 \times 10$ | $5.06 \times 10^2$ | $6.11 \times 10^2$ | $2.27 \times 10$ | $6.00 \times 10^2$ | $2.02 \times 10$ |
| f6 | 600 | $1.35 \times 10$ | $6.00 \times 10^2$ | $6.00 \times 10^2$ | $1.36 \times 10^{-2}$ | $6.00 \times 10^2$ | $1.13 \times 10^{-2}$ |
| f7 | 700 | $1.73 \times 10$ | $7.19 \times 10^2$ | $8.53 \times 10^2$ | $1.68 \times 10$ | $8.60 \times 10^2$ | $1.70 \times 10$ |
| f8 | 800 | $1.44 \times 10$ | $8.07 \times 10^2$ | $9.05 \times 10^2$ | $2.12 \times 10$ | $9.06 \times 10^2$ | $2.27 \times 10$ |
| f9 | 900 | $3.07 \times 10^2$ | $9.00 \times 10^2$ | $9.01 \times 10^2$ | $6.89 \times 10^{-1}$ | $9.04 \times 10^2$ | $1.44 \times 10$ |
| f10 | 1000 | $3.79 \times 10^2$ | $1.40 \times 10^3$ | $6.21 \times 10^3$ | $6.89 \times 10^{-1}$ | $6.05 \times 10^3$ | $5.03 \times 10^2$ |
| f11 | 1100 | $5.24 \times 10$ | $1.10 \times 10^3$ | $1.19 \times 10^3$ | $2.31 \times 10$ | $1.19 \times 10^3$ | $2.90 \times 10$ |
| f12 | 1200 | $4.13 \times 10^6$ | $1.50 \times 10^3$ | $1.56 \times 10^5$ | $1.03 \times 10^5$ | $1.81 \times 10^5$ | $1.48 \times 10^5$ |
| f13 | 1300 | $7.68 \times 10^3$ | $1.31 \times 10^3$ | $3.56 \times 10^3$ | $9.09 \times 10^2$ | $4.01 \times 10^3$ | $1.00 \times 10^3$ |
| f14 | 1400 | $4.18 \times 10^3$ | $1.40 \times 10^3$ | $1.50 \times 10^3$ | $1.03 \times 10$ | $1.50 \times 10^3$ | $1.06 \times 10$ |
| f15 | 1500 | $2.37 \times 10^4$ | $1.50 \times 10^3$ | $1.70 \times 10^3$ | $4.43 \times 10$ | $1.70 \times 10^3$ | $3.95 \times 10$ |
| f16 | 1600 | $1.59 \times 10^2$ | $1.60 \times 10^3$ | $2.45 \times 10^3$ | $2.66 \times 10^2$ | $2.47 \times 10^3$ | $2.14 \times 10^2$ |
| f17 | 1700 | $8.40 \times 10$ | $1.70 \times 10^3$ | $1.85 \times 10^3$ | $6.65 \times 10$ | $1.86 \times 10^3$ | $8.06 \times 10$ |
| f18 | 1800 | $1.79 \times 10^4$ | $1.81 \times 10^3$ | $2.87 \times 10^3$ | $6.86 \times 10^2$ | $2.69 \times 10^3$ | $3.69 \times 10^2$ |
| f19 | 1900 | $3.83 \times 10^4$ | $1.90 \times 10^3$ | $1.99 \times 10^3$ | $1.88 \times 10$ | $1.99 \times 10^3$ | $1.87 \times 10$ |
| f20 | 2000 | $1.08 \times 10^2$ | $2.00 \times 10^3$ | $2.39 \times 10^3$ | $2.58 \times 10$ | $2.22 \times 10^3$ | $1.12 \times 10^2$ |
| f21 | 2100 | $4.78 \times 10$ | $2.23 \times 10^3$ | $2.40 \times 10^3$ | $2.28 \times 10$ | $2.39 \times 10^3$ | $2.51 \times 10$ |
| f22 | 2200 | $5.88 \times 10^2$ | $2.29 \times 10^3$ | $2.30 \times 10^3$ | $1.50 \times 10^{-2}$ | $2.30 \times 10^3$ | $7.44 \times 10^{-3}$ |

**Table 7.** *Cont.*

| | | HFPSO [40] | | DSFS Type-1 Fuzzy Logic | | DSFS Type-2 Fuzzy Logic | |
|---|---|---|---|---|---|---|---|
| **Function** | **fi** | **Mean** | **Std** | **Mean** | **Std** | **Mean** | **Std** |
| f23 | 2300 | $2.87 \times 10$ | $2.61 \times 10^3$ | $2.74 \times 10^3$ | $2.86 \times 10$ | $2.73 \times 10^3$ | $2.26 \times 10$ |
| f24 | 2400 | $1.47 \times 10^2$ | $2.61 \times 10^3$ | $2.91 \times 10^3$ | $3.37 \times 10$ | $2.90 \times 10^3$ | $3.04 \times 10$ |
| f25 | 2500 | $5.02 \times 10$ | $2.90 \times 10^3$ | $2.89 \times 10^3$ | $1.73$ | $2.89 \times 10^3$ | $9.76 \times 10^{-1}$ |
| f26 | 2600 | $3.42 \times 10^2$ | $2.90 \times 10^3$ | $4.28 \times 10^3$ | $5.50 \times 10^2$ | $4.22 \times 10^3$ | $6.59 \times 10^2$ |
| f27 | 2700 | $3.94 \times 10$ | $3.09 \times 10^3$ | $3.22 \times 10^3$ | $8.08$ | $3.22 \times 10^3$ | $7.37$ |
| f28 | 2800 | $1.08 \times 10^2$ | $3.09 \times 10^3$ | $3.21 \times 10^3$ | $1.25 \times 10$ | $3.21 \times 10^3$ | $1.27 \times 10$ |
| f29 | 2900 | $9.40 \times 10$ | $3.16 \times 10^3$ | $3.63 \times 10^3$ | $1.06 \times 10^2$ | $3.59 \times 10^3$ | $1.13 \times 10^2$ |
| f30 | 3000 | $3.75 \times 10^6$ | $3.56 \times 10^3$ | $1.32 \times 10^4$ | $3.29 \times 10^3$ | $1.51 \times 10^4$ | $5.55 \times 10^3$ |

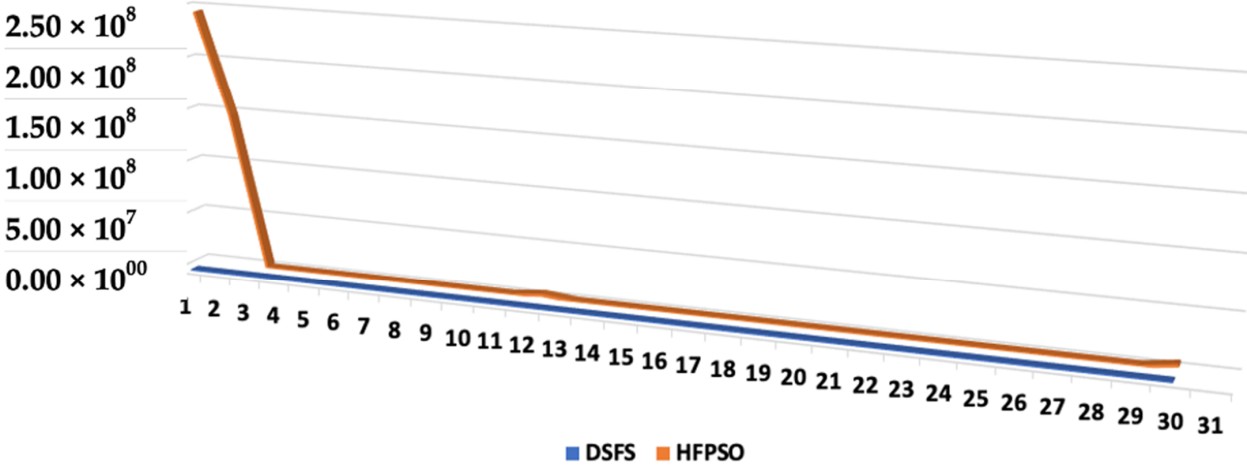

**Figure 3.** DSFS vs. HFPSO Results with 10 dimensions.

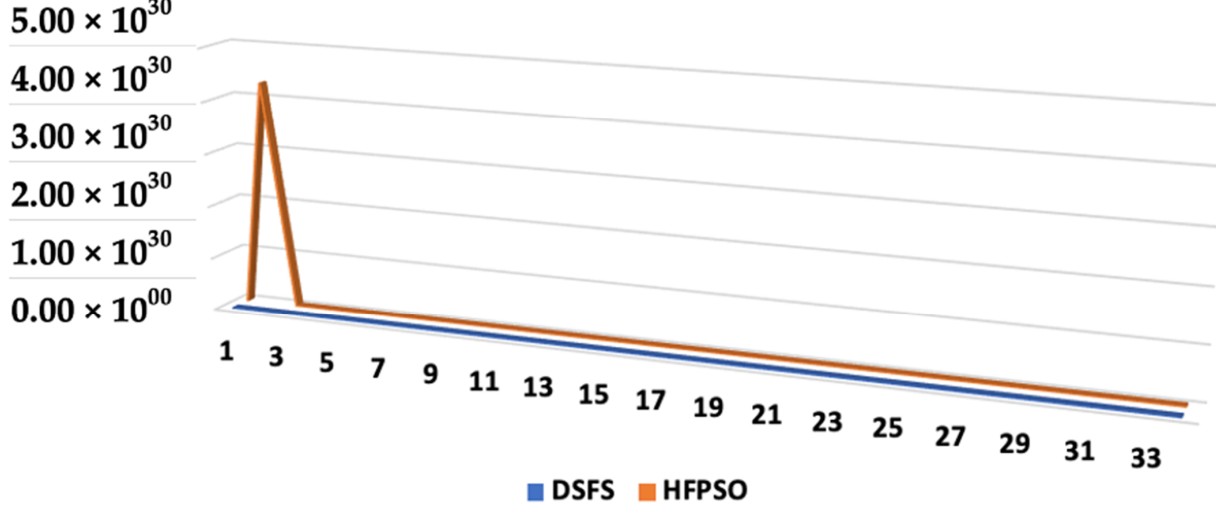

**Figure 4.** Results of the CEC 2017 with 30 dimensions.

On the vertical axis, we can find the values obtained with the DSFS and HFPSO methods, and the horizontal axis shows the corresponding functions. The DSFS approach, indicated by the blue line, has better efficiency than the HFPSO (orange line) because the values are closer to the global optima, as seen in Figure 3.

Figure 4 illustrates the results obtained with the DSFS and HFPSO using 30 dimensions. The vertical axis contains the values reached by the methods when evaluating the CEC 2017 functions and the horizontal axis shows the corresponding function. The orange line is representative of the HFPSO algorithm, where it is observed that, in the first functions, the values were very separate from the optimal ones. Then, for the f1, f2, f3, and f4 functions, the achieved performance can be viewed as unsatisfactory, with results up to $10^{+30}$. On the other hand, the DFSF method obtained results closer to the global optimal, represented by the blue line. We will have to remember that the dimensions are high, and therefore, the performance of the methods is not the same as when using lower dimensions, where the efficiency of the algorithms is much better for each of the functions.

To determine which of the HFPSO or DSFS provided the closest to the optimal result for each function, a statistical comparison was performed using the parametric Z-test. The formula for the Z-test is expressed mathematically in the following fashion:

$$Z = \frac{(\bar{x}_1 - \bar{x}_2) - (\mu_1 - \mu_2)}{\sigma_{\bar{x}_1 - \bar{x}_2}} \tag{13}$$

where $\bar{x}_1 - \bar{x}_2$ is the observed difference in the mean values of the methods, $\mu_1 - \mu_2$ is the expected difference in the mean values of the methods, and $\sigma_{\bar{x}_1 - \bar{x}_2}$ is the standard error of the differences.

As can be seen in Table 8, Column 7, the values obtained by the Z-test provide statistical evidence that the DSFS method using type-1 fuzzy logic is significantly better than HFPSO in the functions with 10 dimensions (bold type indicates the best values).

**Table 8.** Z-test for 10 dimensions.

| Function | fi | HFPSO [40] Mean | Std | DSFS Type-1 Fuzzy Logic Mean | Std | z |
|---|---|---|---|---|---|---|
| f1 | 100 | $9.81 \times 10^8$ | $1.01 \times 10^2$ | $1.01 \times 10^2$ | $4.87 \times 10^{-1}$ | $3.76 \times 10^7$ |
| f2 | 200 | $4.91 \times 10^8$ | $2.00 \times 10^2$ | $2.00 \times 10^2$ | 0.00 | $9.51 \times 10^6$ |
| f3 | 300 | $5.96 \times 10^3$ | $3.00 \times 10^2$ | $3.00 \times 10^2$ | $3.73 \times 10^{-6}$ | $7.31 \times 10$ |
| f4 | 400 | $4.55 \times 10$ | $4.01 \times 10^2$ | $4.01 \times 10^2$ | $7.81 \times 10^{-1}$ | $-3.43$ |
| f5 | 500 | $1.84 \times 10$ | $5.06 \times 10^2$ | $5.06 \times 10^2$ | 2.04 | $-3.73$ |
| f6 | 600 | $1.35 \times 10$ | $6.00 \times 10^2$ | $6.00 \times 10^2$ | $7.21 \times 10^{-8}$ | $-3.79$ |
| f7 | 700 | $1.73 \times 10$ | $7.19 \times 10^2$ | $7.19 \times 10^2$ | 3.45 | $-3.78$ |
| f8 | 800 | $1.44 \times 10$ | $8.07 \times 10^2$ | $8.07 \times 10^2$ | 3.04 | $-3.80$ |
| f9 | 900 | $3.07 \times 10^2$ | $9.00 \times 10^2$ | $9.00 \times 10^2$ | 0.00 | $-2.55$ |
| f10 | 1000 | $3.79 \times 10^2$ | $1.40 \times 10^3$ | $1.40 \times 10^3$ | $1.70 \times 10^2$ | $-2.82$ |
| f11 | 1100 | $5.24 \times 10$ | $1.10 \times 10^3$ | $1.10 \times 10^3$ | $9.31 \times 10^{-1}$ | $-3.69$ |
| f12 | 1200 | $4.13 \times 10^6$ | $1.50 \times 10^3$ | $1.50 \times 10^3$ | $1.04 \times 10^2$ | $1.07 \times 10^4$ |
| f13 | 1300 | $7.68 \times 10^3$ | $1.31 \times 10^3$ | $1.31 \times 10^3$ | 4.03 | $1.88 \times 10$ |
| f14 | 1400 | $4.18 \times 10^3$ | $1.40 \times 10^3$ | $1.40 \times 10^3$ | 1.69 | 7.69 |
| f15 | 1500 | $2.37 \times 10^4$ | $1.50 \times 10^3$ | $1.50 \times 10^3$ | $7.24 \times 10^{-1}$ | $5.73 \times 10$ |
| f16 | 1600 | $1.59 \times 10^2$ | $1.60 \times 10^3$ | $1.60 \times 10^3$ | $2.79 \times 10^{-1}$ | $-3.49$ |
| f17 | 1700 | $8.40 \times 10$ | $1.70 \times 10^3$ | $1.70 \times 10^3$ | 2.68 | $-3.68$ |
| f18 | 1800 | $1.79 \times 10^4$ | $1.81 \times 10^3$ | $1.81 \times 10^3$ | 2.56 | $3.44 \times 10$ |
| f19 | 1900 | $3.83 \times 10^4$ | $1.90 \times 10^3$ | $1.90 \times 10^3$ | $4.15 \times 10^{-1}$ | $7.42 \times 10$ |
| f20 | 2000 | $1.08 \times 10^2$ | $2.00 \times 10^3$ | $2.00 \times 10^3$ | $1.39 \times 10^{-1}$ | $-3.66$ |
| f21 | 2100 | $4.78 \times 10$ | $2.23 \times 10^3$ | $2.23 \times 10^3$ | $5.08 \times 10$ | $-3.79$ |

**Table 8.** *Cont.*

| | | HFPSO [40] | | DSFS Type-1 Fuzzy Logic | | |
|---|---|---|---|---|---|---|
| **Function** | **fi** | **Mean** | **Std** | **Mean** | **Std** | **z** |
| f22 | 2200 | $5.88 \times 10^2$ | $2.29 \times 10^3$ | $2.29 \times 10^3$ | $2.17 \times 10$ | $-2.88$ |
| f23 | 2300 | $2.87 \times 10$ | $2.61 \times 10^3$ | $2.61 \times 10^3$ | $2.80$ | $-3.83$ |
| f24 | 2400 | $1.47 \times 10^2$ | $2.61 \times 10^3$ | $2.61 \times 10^3$ | $1.22 \times 10^2$ | $-3.65$ |
| f25 | 2500 | $5.02 \times 10$ | $2.90 \times 10^3$ | $2.90 \times 10^3$ | $1.08 \times 10$ | $-3.81$ |
| f26 | 2600 | $3.42 \times 10^2$ | $2.90 \times 10^3$ | $2.90 \times 10^3$ | $2.15 \times 10^{-10}$ | $-3.42$ |
| f27 | 2700 | $3.94 \times 10$ | $3.09 \times 10^3$ | $3.09 \times 10^3$ | $1.97$ | $-3.82$ |
| f28 | 2800 | $1.08 \times 10^2$ | $3.09 \times 10^3$ | $3.09 \times 10^3$ | $4.20 \times 10$ | $-3.74$ |
| f29 | 2900 | $9.40 \times 10$ | $3.16 \times 10^3$ | $3.16 \times 10^3$ | $1.01 \times 10$ | $-3.76$ |
| f30 | 3000 | $3.75 \times 10^6$ | $3.56 \times 10^3$ | $3.56 \times 10^3$ | $2.30 \times 10^2$ | $4.08 \times 10^3$ |

Table 9 describes in Column 7 the results when applying the Z-test to the methods, compared to DSFS with type-1 fuzzy logic vs. HFPSO using 30 dimensions. Again, the proposed method is significantly better. For example, in Row 2, for the f2 function, the Z value provides sufficient statistical evidence that the proposed DSFS method is better.

**Table 9.** Results Z-test for 30 dimensions.

| | | HFPSO [40] | | DSFS Type-1 Fuzzy Logic | | |
|---|---|---|---|---|---|---|
| **Function** | **fi** | **Mean** | **Std** | **Mean** | **Std** | **z** |
| f1 | 100 | $9.81 \times 10^8$ | $1.01 \times 10^2$ | $3.49 \times 10^3$ | $2.76 \times 10^3$ | $1.95 \times 10^6$ |
| f2 | 200 | $4.91 \times 10^8$ | $2.00 \times 10^2$ | $3.06 \times 10^{16}$ | $7.11 \times 10^{16}$ | $-2.36$ |
| f3 | 300 | $5.96 \times 10^3$ | $3.00 \times 10^2$ | $8.40 \times 10^3$ | $3.95 \times 10^3$ | $-3.37$ |
| f4 | 400 | $4.55 \times 10$ | $4.01 \times 10^2$ | $4.87 \times 10^2$ | $3.56 \times 10$ | $-6.01$ |
| f5 | 500 | $1.84 \times 10$ | $5.06 \times 10^2$ | $6.11 \times 10^2$ | $2.27 \times 10$ | $-6.41$ |
| f6 | 600 | $1.35 \times 10$ | $6.00 \times 10^2$ | $6.00 \times 10^2$ | $1.36 \times 10^{-2}$ | $-5.35$ |
| f7 | 700 | $1.73 \times 10$ | $7.19 \times 10^2$ | $8.53 \times 10^2$ | $1.68 \times 10$ | $-6.36$ |
| f8 | 800 | $1.44 \times 10$ | $8.07 \times 10^2$ | $9.05 \times 10^2$ | $2.12 \times 10$ | $-6.04$ |
| f9 | 900 | $3.07 \times 10^2$ | $9.00 \times 10^2$ | $9.01 \times 10^2$ | $6.89 \times 10^{-1}$ | $-3.61$ |
| f10 | 1000 | $3.79 \times 10^2$ | $1.40 \times 10^3$ | $6.21 \times 10^3$ | $6.89 \times 10^{-1}$ | $-2.28 \times 10$ |
| f11 | 1100 | $5.24 \times 10$ | $1.10 \times 10^3$ | $1.19 \times 10^3$ | $2.31 \times 10$ | $-5.66$ |
| f12 | 1200 | $4.13 \times 10^6$ | $1.50 \times 10^3$ | $1.56 \times 10^5$ | $1.03 \times 10^5$ | $2.11 \times 10^2$ |
| f13 | 1300 | $7.68 \times 10^3$ | $1.31 \times 10^3$ | $3.56 \times 10^3$ | $9.09 \times 10^2$ | $1.42 \times 10$ |
| f14 | 1400 | $4.18 \times 10^3$ | $1.40 \times 10^3$ | $1.50 \times 10^3$ | $1.03 \times 10$ | $1.05 \times 10$ |
| f15 | 1500 | $2.37 \times 10^4$ | $1.50 \times 10^3$ | $1.70 \times 10^3$ | $4.43 \times 10$ | $8.03 \times 10$ |
| f16 | 1600 | $1.59 \times 10^2$ | $1.60 \times 10^3$ | $2.45 \times 10^3$ | $2.66 \times 10^2$ | $-7.74$ |
| f17 | 1700 | $8.40 \times 10$ | $1.70 \times 10^3$ | $1.85 \times 10^3$ | $6.65 \times 10$ | $-5.69$ |
| f18 | 1800 | $1.79 \times 10^4$ | $1.81 \times 10^3$ | $2.87 \times 10^3$ | $6.86 \times 10^2$ | $4.25 \times 10$ |
| f19 | 1900 | $3.83 \times 10^4$ | $1.90 \times 10^3$ | $1.99 \times 10^3$ | $1.88 \times 10$ | $1.05 \times 10^2$ |
| f20 | 2000 | $1.08 \times 10^2$ | $2.00 \times 10^3$ | $2.39 \times 10^3$ | $2.58 \times 10$ | $-6.25$ |
| f21 | 2100 | $4.78 \times 10$ | $2.23 \times 10^3$ | $2.40 \times 10^3$ | $2.28 \times 10$ | $-5.78$ |
| f22 | 2200 | $5.88 \times 10^2$ | $2.29 \times 10^3$ | $2.30 \times 10^3$ | $1.50 \times 10^{-2}$ | $-4.09$ |
| f23 | 2300 | $2.87 \times 10$ | $2.61 \times 10^3$ | $2.74 \times 10^3$ | $2.86 \times 10$ | $-5.69$ |
| f24 | 2400 | $1.47 \times 10^2$ | $2.61 \times 10^3$ | $2.91 \times 10^3$ | $3.37 \times 10$ | $-5.80$ |
| f25 | 2500 | $5.02 \times 10$ | $2.90 \times 10^3$ | $2.89 \times 10^3$ | $1.73$ | $-5.36$ |
| f26 | 2600 | $3.42 \times 10^2$ | $2.90 \times 10^3$ | $4.28 \times 10^3$ | $5.50 \times 10^2$ | $-7.31$ |
| f27 | 2700 | $3.94 \times 10$ | $3.09 \times 10^3$ | $3.22 \times 10^3$ | $8.08$ | $-5.64$ |
| f28 | 2800 | $1.08 \times 10^2$ | $3.09 \times 10^3$ | $3.21 \times 10^3$ | $1.25 \times 10$ | $-5.50$ |
| f29 | 2900 | $9.40 \times 10$ | $3.16 \times 10^3$ | $3.63 \times 10^3$ | $1.06 \times 10^2$ | $-6.13$ |
| f30 | 3000 | $3.75 \times 10^6$ | $3.56 \times 10^3$ | $1.32 \times 10^4$ | $3.29 \times 10^3$ | $4.22 \times 10^3$ |

## 6. Conclusions

In conclusion, we can propose efficient stochastic methods, such as metaheuristics, that include basic algorithms for global stochastic optimization, such as the random search, which helps with the dispersion of the particles to reach the global optimum and avoid local stagnation. In this study, an improvement was made to a recent method that has been used in several optimization problems. This algorithm has two important processes for its operation: the diffusion and the updating process, which make up the most important parts of the stochastic fractal search (SFS). This algorithm has been dynamically analyzed and improved with a fuzzy controller, which controls the diffusion of particles and diversity by iteration, making this a better optimization method, as has been observed in the results obtained, compared to others.

The experimentation was carried out first with the comparison of the results of the dynamic stochastic fractal search method, adjusted with both kinds of fuzzy logic, where we found that the values were very similar for all the CEC 2017 evaluation functions. After that, a comparison with the combination of the hybrid firefly algorithm and particle swarm optimization (HFPSO) was also done, showing a significant statistical advantage for the proposed DSFS method of this paper. In addition, since HFPSO was better than the FA, PSO, HPSOFF, and FFPSO [39], the proposed DSFS is also better than these five metaheuristic optimization algorithms.

It is shown that the improvement applied to the method was satisfactory because of its efficiency and optimization performance. This improvement was made by adding the iteration and diversity equations to help the chaotic stochastic movement of the particles. In addition, it was adjusted with a fuzzy inference controller. Finally, it can be concluded that the combination of stochastic metaheuristics with fuzzy logic can generate good results for the efficiency and improvement of the optimization algorithms, as was developed in this study. In future work, we plan to apply the proposed DFSF to real-world problems in different areas of application.

**Author Contributions:** Conceptualization, M.L.L. and O.C.; methodology, P.M.; software, F.V.; validation, J.S., M.L.L. and O.C.; formal analysis, J.S.; investigation, M.L.L.; resources, P.M.; data curation, M.L.L.; writing—original draft preparation, M.L.L.; writing—review and editing, O.C.; visualization, F.V.; supervision, O.C.; project administration, P.M.; funding acquisition, P.M. All authors have read and agreed to the published version of the manuscript.

**Funding:** This research received no external funding.

**Institutional Review Board Statement:** Not applicable.

**Informed Consent Statement:** Not applicable.

**Data Availability Statement:** Not applicable.

**Conflicts of Interest:** The authors declare no conflict of interest.

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
