# Peer review of "A New Approach for Dynamic Stochastic Fractal Search with Fuzzy Logic for Parameter Adaptation"

_fractalfract, doi:10.3390/fractalfract5020033_

Round 1

Reviewer 1 Report

the input information used in the fuzzy system to control the values
 of diffusion.

The paper looks interesting but need minor revision. I suggest to consider the following minor correction:
1- The Abstract must be revised. Some sentences might shifted to the Introduction.
2- The Title of Section 2 must be edited. I suggest "Materials and Methods for Stochastic Fractal Search (SFS)".
3- Change "Stochastic Fractal Search (SFS)" to "stochastic fractal search (SFS)". Also do it for "Dynamic Stochastic Fractal Search (DFSF)".
4-Observe the punctuation at the end of the  equations. See for example Eqs. (10), (11) and (12).

Author Response

The paper looks interesting but need minor revision. I suggest to consider the following minor correction:
1- The Abstract must be revised. Some sentences might shifted to the Introduction.

R: Thanks for your observation. The first sentence of the Abstract has been moved to the introduction. We have also improved both the Abstract and the Introduction.

2- The Title of Section 2 must be edited. I suggest "Materials and Methods for Stochastic Fractal Search (SFS)".

R: Thank you for your comment. The suggested change for the title of Section 2 has been made.

3- Change "Stochastic Fractal Search (SFS)" to "stochastic fractal search (SFS)". Also do it for "Dynamic Stochastic Fractal Search (DFSF)".

R: Thanks for the suggested correction. The suggested changes have been applied in the new version of the manuscript.

4-Observe the punctuation at the end of the equations. See for example Eqs. (10), (11) and (12).

R: Thanks for your observation. The suggested changes have been made in the new version of the manuscript.

Reviewer 2 Report

This is a paper that tries to use fuzzy set theory with fractal methods. Not something original in the sense that someone else has done something similar already but not exactly what the authors describe. Therefore, I think it is worth publishing it.

Author Response

This is a paper that tries to use fuzzy set theory with fractal methods. Not something original in the sense that someone else has done something similar already but not exactly what the authors describe. Therefore, I think it is worth publishing it.

R: We thank the reviewer for the positive comments on our paper.

Reviewer 3 Report

The paper aims to examine an exciting issue of both theoretical and application-based potential. It seems to be noteworthy and potentially interesting for the journal audience.  The size of the computation is impressive. 

However, some places of the paper seem to require either a more profound revision or some complementary methodological explanation. 

  1. Although SFS and DSFS are explained in detail and a novelty of the paper is indicated in 'Introduction', it is not clear how fuzzy inference supports the SFS/DSFS to guarantee the computation results collected in the tables. /The reader can only suppose that the formulae (9)-(12) might play some role in the computation, but it is not explicitly shown. As a consequence, Popper's falsifiability criterion is hardly applicable to the paper's thesis./  
  2. It is strongly recommended to elucidate the paper's motivation clearly. It is carefully hidden in the initial state-of-the-art' description and in the paper's objectives. 
  3. In my opinion, the formulae (1)-(4) are introduced in an encyclopedic way. I would recommend modifying the presentation style towards a systematic derivation of the formulae (even as did in (5)-(8)).  
  4. Some sentences are hardly readable, for example, the sentence in lines 65-76, 147-153, and others. 
  5. The punctuation should be improved in many places (especially before: 'a', 'the', 'this'). 
  6. Since the idea of metaheuristics seems to be crucial for a philosophical or methodological foundation of the paper's analysis, this concept, its semantic field, and its role in the paper's considerations should be explained better. /Especially that 'multi-metaheuristic models' are even mentioned in line 51/.  
  7. I would recommend describing the content of the tables in a more detailed way (for example, by describing the first rows of them) to give a reader a chance to comprehend the author's computations. 

Author Response

The paper aims to examine an exciting issue of both theoretical and application-based potential. It seems to be noteworthy and potentially interesting for the journal audience.  The size of the computation is impressive. 

R: We thank the reviewer for the positive comments on our paper.

However, some places of the paper seem to require either a more profound revision or some complementary methodological explanation. 

  1. Although SFS and DSFS are explained in detail and a novelty of the paper is indicated in 'Introduction', it is not clear how fuzzy inference supports the SFS/DSFS to guarantee the computation results collected in the tables. /The reader can only suppose that the formulae (9)-(12) might play some role in the computation, but it is not explicitly shown. As a consequence, Popper's falsifiability criterion is hardly applicable to the paper's thesis./  

       R: Thank you for your comment. We have made every possible effort to explain the role that fuzzy logic plays in the proposed approach.

2. It is strongly recommended to elucidate the paper's motivation clearly. It is carefully hidden in the initial state-of-the-art' description and in the paper's objectives.

      R: Thanks for your observation. In the new version of the paper we have explained in more detail the motivation of the paper.

3. In my opinion, the formulae (1)-(4) are introduced in an encyclopedic way. I would recommend modifying the presentation style towards a systematic derivation of the formulae (even as did in (5)-(8)).  

      R: Thanks for your comments. We have tried to improve the presentation of the equations as much as possible.

4. Some sentences are hardly readable, for example, the sentence in lines 65-76, 147-153, and others. 

R: Thank you for pointing out this issue with the paper. We have made every effort to improve the readability of the sentences.

5. The punctuation should be improved in many places (especially before: 'a', 'the', 'this').

R: Thank you for pointing out this issue with the paper. We have improved the punctuation in the paper, as suggested by the reviewer.

6. Since the idea of metaheuristics seems to be crucial for a philosophical or methodological foundation of the paper's analysis, this concept, its semantic field, and its role in the paper's considerations should be explained better. /Especially that 'multi-metaheuristic models' are even mentioned in line 51/.  

     R: Thanks for your observation. We have made every possible effort to explain the role that the metaheuristic concept plays in the proposed approach.

7. I would recommend describing the content of the tables in a more detailed way (for example, by describing the first rows of them) to give a reader a chance to comprehend the author's computations. 

    R: Thank you for your comment. We have tried to improve the description of the tables to offer the readers the opportunity to understand better the results.